# Tensor-Decomposed RNNs for Marked Temporal Point Processes

**Timothy Mulumba**  *timothy.mulumba@nyu.edu*
*Division of Engineering*
*New York University Abu Dhabi, Abu Dhabi, United Arab Emirates*
*AI & Robotics Kampala (ARK) Lab, Kampala, Uganda*

**Reviewed on OpenReview:** *https: // openreview. net/ forum? id= YOup92GcjF*

## Abstract

We study parameter-efficient neural Marked Temporal Point Processes (MTPPs) for high-dimensional mark and exogenous feature spaces. Building on tensor-train (TT) factorization of recurrent kernels, we propose *mark-aware TT shaping* that aligns TT cores with known multi-way domain structure (e.g., asset/venue/side in finance). We provide a conditional intensity function-consistent training recipe and evaluate both accuracy and calibration (mark reliability and time-rescaling diagnostics). Across finance and public MTPP benchmarks, TT-compressed RNNs reduce parameters by 40–70% while matching dense baselines and remaining competitive with attention-based and state-space models. Capacity-controlled and shaping ablations show that the observed calibration gains are not solely due to smaller parameter count: parameter-matched dense models improve calibration partly, while mark-aware TT ordering further improves likelihood, expected calibration error (ECE), and time-rescaling diagnostics over random or unstructured TT orderings.

## 1 Introduction

Modeling high-frequency event streams with multiple types (or *marks*) is a central problem in application-driven machine learning, especially in domains such as financial markets, image and video processing, and social network analysis (Du et al., 2015; Xu & Zha, 2017; Schein et al., 2015). In the financial domain, for instance, practitioners often track requests for quotes (RFQs), limit-order book events, or trades in a continuous-time framework where arrival patterns are inherently interdependent. These applications demand not only accurate short-term forecasting but also robust cross-excitation effects among event types.

A Marked Temporal Point Process (MTPP) (Daley & Vere-Jones, 2003) can explicitly capture how each event's arrival influences future arrivals of multiple types. These Marked Temporal Point Processes (MTPPs) are well-suited to modeling sequences of events in continuous time, where each event has both a time and a mark (type). In settings such as financial markets, high-frequency limit-order books, or requests for quotes (RFQs), capturing cross-excitation among multiple event types is essential (Boyd et al., 2020; Du et al., 2016; Bhattacharjya et al., 2018). Neural variants of MTPPs extend classic self-exciting processes by encoding history in a hidden state of a recurrent neural network (RNN), but in high-dimensional settings the input-to-hidden and hidden-to-hidden maps can dominate the parameter budget and memory footprint (Tjandra et al., 2017; Yang et al., 2017). Specifically, these neural approaches incorporate RNNs to encode the event history $(x_t, h_t)$ in a hidden state, from which the conditional intensity function $\lambda_x(t)$ can be parameterized (Zhang et al., 2020; Türkmen et al., 2019). *Tensor-train decomposition* (Oseledets, 2011) offers an elegant way to reduce the parameter count by factorizing large weight matrices into products of smaller core tensors.

**Contributions.** Our contribution is intentionally design- and empirics-focused. We provide: (i) a likelihood-preserving way to replace the large dense maps inside RNN-based MTPPs with TT layers while keeping the conditional intensity heads unchanged; (ii) a mark-aware TT shaping strategy that orders TT cores

according to multi-way mark/exogenous structure; (iii) controlled ablations against random and unstructured TT factorizations, parameter-matched dense LSTMs, and regularized dense LSTMs; and (iv) an evaluation protocol that reports likelihood, mark accuracy, parameter/FLOP accounting, wall-clock timing, mark calibration, time-rescaling diagnostics, and prediction-interval coverage.

**Why TT for RNN-based MTPPs (and not only Transformers)?** RNN-based neural point processes remain a standard continuous-time modeling approach (Du et al., 2016; Mei & Eisner, 2017) with a clean intensity-based likelihood and time-rescaling diagnostics. TT can be inserted into this pipeline by replacing only the large linear maps inside the sequence model, keeping the conditional intensity and loss unchanged (Sec. 4). We additionally compare against strong attention and state-space baselines (THP/SAHP and S4/Mamba; Sec. 7); applying TT to Transformer layers is an appealing extension, but is orthogonal to our goal of isolating how tensor compression affects efficiency and calibration in continuous-time MTPPs.

**Paper Organization.** We first review related work in Section 2. We then review MTPPs and RNN memory in Sections 3–4. Section 5 presents the tensor-train decomposition and how it replaces large dense transformations in an RNN cell. Next, we present experimental results in Sections 6–7, including ablations. Finally, in Section 8, we summarize insights and discuss future directions.

## 2 Related Work

**Marked Temporal Point Processes.** Classical self-exciting processes (e.g., Hawkes processes (Hawkes, 1971; Ogata, 1988)) have a long history in finance, biological processes and social sciences (Bacry et al., 2015; Hawkes, 2022; Laub et al., 2015). They capture how past events raise or dampen future intensities. Extending Hawkes to neural architectures has led to advanced MTPP models that encode the event history with recurrent (Du et al., 2016; Mei & Eisner, 2017) or attention-based mechanisms (Zuo et al., 2020; Zhang et al., 2020). Our work follows this RNN-based lineage but differs by applying tensor decomposition to manage high input dimension. State-space models (SSMs) have also found recent success as alternatives to RNNs and transformers, offering efficient linear-time training and inference (Gao et al., 2024; Chang et al., 2024).

**TT-compressed RNNs.** Low-rank tensor approximations (CP decomposition, Tucker, tensor-train) reduce parameters in large weight matrices (Oseledets, 2011). Prior work compresses input and recurrent matrices via tensor-train and related decompositions for speech/video and sequence tasks (Novikov et al., 2015; Tjandra et al., 2017; Yang et al., 2017), and for finance forecasting (Xu et al., 2021). We do not claim TT-RNN compression itself as novel. The distinction in this paper is the MTPP-specific integration: the TT layer replaces only the RNN maps, while the conditional intensity, survival integral, and event-time likelihood remain unchanged. This lets us evaluate TT compression using point-process diagnostics such as time-rescaling and mark reliability. We further study whether semantic ordering of TT cores matters for marked event data through matched-rank shaping ablations, which is not addressed by generic TT-RNN compression work. We also benchmark TT against tensor ring and Tucker/CP variants to isolate design trade-offs (Pan et al., 2019), and make systematic comparisons to modern attention and state-space baselines (Zuo et al., 2020; Zhang et al., 2020; Gu et al., 2021; Gu & Dao, 2023).

**Reinforcement Learning (RL) Approaches.** By formulating different objectives, RL allows for bypassing the computational issues related to evaluating the integral in the MTTP likelihood (Li et al., 2018; Upadhyay et al., 2018; Zhu et al., 2021).

**Applications in Finance.** In real-world financial data, events arrive at irregular intervals, and multiple instrument types or currency pairs may interact (Bacry et al., 2015; Laub et al., 2015). Neural MTPPs can capture such cross-asset excitation but at the expense of parameter blow-up when the dimensionality of marks or engineered features is large. Traditional domain-specific solutions (e.g., parametric Hawkes with carefully chosen kernels or GARCH-like processes for volatility become unwieldy when the number of event types grows (Engle, 2000)). We bridge these approaches by providing a flexible RNN-based MTPP that remains computationally feasible even for dozens of assets or features, thanks to TT-based compression.

**Non-ML and Baseline Approaches.** Outside the ML literature, many financial firms rely on simpler methods, such as moving averages or partially parametric hazard functions, to forecast demand or trade arrivals. While these can be effective in narrow domains, they lack the adaptive memory of modern RNN-based MTPPs. Hence, we also compare to baseline ML methods (dense RNN) and discuss performance trade-offs to highlight the value of TT-based compression for large-scale real-world settings.

## 3 Background on Marked Temporal Point Processes

We model an event stream on $[0, T]$ as a sequence $\mathcal{D} = \{(t_i, x_i)\}_{i=1}^N$ with strictly increasing times $0 < t_1 < \cdots < t_N \leq T$ and marks $x_i \in \{1, \ldots, K\}$. Let $\mathcal{H}_t = \{(t_j, x_j) : t_j < t\}$ denote the history up to $t$. A marked temporal point process (MTPP) is characterized by mark–specific conditional intensities

$$
\begin{aligned}
\lambda_k(t \mid \mathcal{H}_t)\, dt \;=\; & \Pr\{\text{event of type } k \text{ in } [t, t+dt) \mid \mathcal{H}_t\}, \\
& k = 1, \ldots, K.
\end{aligned}
\tag{1}
$$

with total intensity $\lambda(t \mid \mathcal{H}_t) = \sum_{k=1}^K \lambda_k(t \mid \mathcal{H}_t)$. Given $\mathcal{H}_{t_i}$, the joint density of the next event's time $t_{i+1}$ and mark $x_{i+1}$ has the hazard–survival form:

$$
\begin{aligned}
f(t_{i+1}, x_{i+1} \mid \mathcal{H}_{t_i}) \;=\; & \lambda_{x_{i+1}}(t_{i+1} \mid \mathcal{H}_{t_{i+1}}) \\
& \times \exp\Big( -\int_{t_i}^{t_{i+1}} \lambda(s \mid \mathcal{H}_s)\, ds \Big).
\end{aligned}
\tag{2}
$$

Equivalently, this factorizes into a time component and a (potentially time–dependent) mark distribution:

$$
\begin{aligned}
f(t_{i+1} \mid \mathcal{H}_{t_i}) =& \lambda(t_{i+1} \mid \mathcal{H}_{t_{i+1}}) \\
& \times \exp\Big( -\int_{t_i}^{t_{i+1}} \lambda(s \mid \mathcal{H}_s)\, ds \Big),
\end{aligned}
\tag{3}
$$

$$
p(x_{i+1} = k \mid t_{i+1}, \mathcal{H}_{t_{i+1}}) = \frac{\lambda_k(t_{i+1} \mid \mathcal{H}_{t_{i+1}})}{\lambda(t_{i+1} \mid \mathcal{H}_{t_{i+1}})}.
\tag{4}
$$

For a realized sequence $\mathcal{D}$, the (negative) log-likelihood (NLL) on $[0, T]$ is

$$
\mathcal{L}_{\text{TPP}}(\theta) \;=\; -\sum_{i=1}^N \log \lambda_{x_i}(t_i \mid \mathcal{H}_{t_i}) \;+\; \int_0^T \lambda(s \mid \mathcal{H}_s)\, ds.
\tag{5}
$$

## 4 RNN Modeling of Event Sequences → Conditional Intensity

**Hidden-state dynamics.** Let $h_i \in \mathbb{R}^H$ summarize history at the $i$-th event time $t_i$. We update only at event times:

$$
h_i \;=\; \text{RNN}_\theta\big(h_{i-1}, \phi(x_i), \psi(z_i), \varphi(\Delta t_i)\big),
$$

$$
\Delta t_i = t_i - t_{i-1}. \tag{6}
$$

where $\phi(\cdot)$ encodes the mark, $z_i$ denotes optional exogenous features, and $\varphi(\Delta t)$ is a time embedding (e.g., Fourier or radial basis). We replace all dense linear maps in $\text{RNN}_\theta$ (input and recurrent) by TT-layers, keeping the conditional intensity and the likelihood unchanged (Section 5).

**Two-head parameterization of the MTPP.** Conditioned on $h_i$ and the elapsed time $\tau = t - t_i$, we parameterize:

**(mark head)**

$$
\eta(h_i) = W_\eta h_i + b_\eta \in \mathbb{R}^K, \qquad W_\eta \in \mathbb{R}^{K \times H},\ b_\eta \in \mathbb{R}^K,
$$

$$
\pi_k(\tau, h_i) = \text{softmax}_k\big(\eta(h_i) + a_k\, \tau\big), \qquad a \in \mathbb{R}^K,
\tag{7}
$$

**(time head)**

$$\lambda(\tau, h_i) = \text{softplus}\Big(w_\lambda^\top [\, h_i;\, \phi_\tau(\tau)\,] + b_\lambda\Big), \qquad w_\lambda \in \mathbb{R}^{H+d_\tau}. \tag{8}$$

Giving mark-specific intensities $\lambda_k(t \mid \mathcal{H}_t) = \lambda(\tau, h_i) \cdot \pi_k(\tau, h_i)$, Eqs. 2–5 apply. Here $a_k\tau$ is a mark-specific elapsed-time drift in the logits. This is a parsimonious time-dependent mark model: it allows the relative probability of marks to change between events, but restricts that dependence to be linear in $\tau$. We use this form for stability and interpretability; replacing $a_k\tau$ with a nonlinear function $g_k(\tau, h_i)$ is compatible with the likelihood factorization but is outside the scope of the present compression study. We adopt either an RMTPP-style closed form for $\lambda$ or a general neural hazard integrated by Gauss–Legendre quadrature (Du et al., 2016; Mei & Eisner, 2017).

**From hidden state to likelihood.** With $\lambda_k(\cdot)$ defined, the sequence NLL is

$$\mathcal{L}_{\text{TPP}}(\theta) = -\sum_{i=1}^{N} \log \lambda_{x_i}(t_i \mid \mathcal{H}_{t_i}) + \int_0^T \lambda(s \mid \mathcal{H}_s)\, ds. \tag{9}$$

Under the decomposition $\lambda_k = \lambda \cdot \pi_k$, Eq. 9 splits into a time term and a mark term:

$$\begin{aligned}
\mathcal{L}_{\text{TPP}} = &-\sum_{i=1}^{N} \log \lambda(t_i \mid \mathcal{H}_{t_i}) \;+\; \int_0^T \lambda(s \mid \mathcal{H}_s)\, ds \\
&- \sum_{i=1}^{N} \log \pi_{x_i}(t_i \mid \mathcal{H}_{t_i})
\end{aligned} \tag{10}$$

which enables separate calibration analysis of marks and times.

**Time-rescaling theorem and PIT diagnostics.** Define rescaled increments $w_i = \int_{t_{i-1}}^{t_i} \lambda(s \mid \mathcal{H}_s)\, ds$. Under a correctly specified model, $\{w_i\}$ are i.i.d. Exp(1); equivalently, $u_i = 1 - e^{-w_i}$ are i.i.d. Uniform$(0,1)$ (Ogata, 1988). We evaluate goodness-of-fit using PIT histograms and a one-sample KS test (Brown et al., 2002).

**Theorem 4.1** (Time-rescaling). *Let $\{(t_i, x_i)\}$ be generated by an MTPP with (total) CIF $\lambda(t \mid \mathcal{H}_t)$. Then $w_i = \int_{t_{i-1}}^{t_i} \lambda(s \mid \mathcal{H}_s)\, ds$ are i.i.d. Exp(1). Consequently $u_i = 1 - \exp(-w_i)$ are i.i.d. Uniform$(0,1)$.*

**From TT-RNN to calibrated MTPP.** Our design uses TT to compress only the linear maps inside the RNN (input and recurrent), leaving the continuous-time head (Eqs. 7–8) and the likelihood (Eq. 9) unchanged. This separation lets us retain neural MTPP flexibility, measure efficiency via parameter counts and wall-clock, and evaluate calibration with time-rescaling and mark reliability.

## 5 Tensor-Train Decomposition for RNN Kernels

When the feature dimension is large, the RNN's weight matrices ($W_{ih}$ from inputs to hidden, and $W_{hh}$ from hidden to hidden) can dominate the parameter budget. A tensor-train factorization (TT) replaces each large matrix with products of smaller core tensors (Oseledets, 2011).

### 5.1 Definition

Let $A \in \mathbb{R}^{M \times N}$ be a weight matrix with $M = \prod_{k=1}^{d} p_k$ and $N = \prod_{k=1}^{d} q_k$. We reshape row and column indices into multi-indices:

$$\begin{aligned}
i &\leftrightarrow (i_1, \ldots, i_d), \quad i_k \in \{1, \ldots, p_k\}, \\
j &\leftrightarrow (j_1, \ldots, j_d), \quad j_k \in \{1, \ldots, q_k\}.
\end{aligned}$$

A TT factorization represents $A$ as a contraction of cores $\{G_k\}_{k=1}^{d}$ with $G_k \in \mathbb{R}^{p_k \times q_k \times r_{k-1} \times r_k}$ and ranks $r_0 = r_d = 1$:

$$A_{ij} = \sum_{\alpha_1=1}^{r_1} \cdots \sum_{\alpha_{d-1}=1}^{r_{d-1}} \prod_{k=1}^{d} G_k[i_k, j_k, \alpha_{k-1}, \alpha_k].$$

---
**Algorithm 1** Forward pass of a TT-matrix layer

---
1: **Input:** mini-batch $\mathbf{x} \in \mathbb{R}^{B \times D_{\text{in}}}$, with $D_{\text{in}} = \prod_{k=1}^{d} q_k$; TT cores $G_k \in \mathbb{R}^{p_k \times q_k \times r_{k-1} \times r_k}$ with $r_0 = r_d = 1$; optional bias $\mathbf{b} \in \mathbb{R}^{D_{\text{out}}}$, $D_{\text{out}} = \prod_{k=1}^{d} p_k$.
2: **Output:** $\mathbf{y} \in \mathbb{R}^{B \times D_{\text{out}}}$.
3: Reshape $X \leftarrow \text{reshape}(\mathbf{x}, B, q_1, \ldots, q_d)$.
4: Initialize $C_0[b, \alpha_0, j_1, \ldots, j_d] \leftarrow X[b, j_1, \ldots, j_d]$ with singleton rank index $\alpha_0 = 1$.
5: **for** $k = 1$ to $d$ **do**
6:     Contract one input mode and one TT-rank index:

$$C_k[b, \alpha_k, i_1, \ldots, i_k, j_{k+1}, \ldots, j_d] \leftarrow \sum_{\alpha_{k-1}=1}^{r_{k-1}} \sum_{j_k=1}^{q_k} C_{k-1}[b, \alpha_{k-1}, i_1, \ldots, i_{k-1}, j_k, \ldots, j_d] \, G_k[i_k, j_k, \alpha_{k-1}, \alpha_k].$$

7: **end for**
8: Reshape $Y \leftarrow \text{reshape}(C_d[b, 1, i_1, \ldots, i_d], B, D_{\text{out}})$.
9: **return** $Y + \mathbf{b}$ if bias is used; otherwise return $Y$.

---

Table 1: Mark-aware TT factor shapes (examples).

| Dataset | Input reshape $M = \prod m_g$ | Notes |
|---|---|---|
| LOBSTER | $m_{\text{asset}} \times m_{\text{venue}} \times m_{\text{side}} \times m_{\text{rest}}$ | couple asset–venue cores |
| FX RFQ | $m_{\text{pair}} \times m_{\text{calendar}} \times m_{\text{rest}}$ | keep calendar core small |
| StackOverflow | $m_{\text{family}} \times m_{\text{subtype}} \times m_{\text{rest}}$ | 22 marks $\rightarrow$ families |
| MIMIC | $m_{\text{system}} \times m_{\text{code}} \times m_{\text{rest}}$ | CCS/ICD-style groupings |

## 5.2 TT-based RNN Cell

To incorporate TT factorization, we reshape $x_t$ into $(B, p_1, p_2, \ldots)$, learn TT cores $\{G_k\}$, and replace dense maps by TT layers that contract across the cores. Likewise for the recurrent transformation on $h_{t-1}$. Algorithm 1 summarizes a TT layer's forward pass.

The TT layer is a linear map. Nonlinearities are applied outside this layer by the surrounding RNN cell (e.g., sigmoid/tanh gates in an LSTM). If an input or output dimension does not factor exactly into the chosen modes, we pad to the nearest product and crop the output after the linear map. Padding is included in the parameter-count table.

**Mark-aware TT shaping.** Many MTPP settings admit a natural grouping of marks and exogenous features. We exploit this structure by reshaping the input dimension $M$ into a product of factors before applying TT, and by ordering TT cores so that adjacent cores align with semantically coupled groups. Table 1 summarizes the reshapes used in our experiments.

**Parameter and FLOP accounting.** For a TT layer replacing a matrix $W \in \mathbb{R}^{M \times D}$ (with $M = \prod p_k$ and $D = \prod q_k$), the parameter count is $\sum_k p_k q_k r_{k-1} r_k$ (plus bias). For a single-layer dense LSTM with input size $M$ and hidden size $H$, the core parameter count is $4H(M + H) + 4H$.

Table 2: Parameter breakdown for the representative LOBSTER setting ($H = 128$). RNN-map compression is larger than full-model compression because embeddings and conditional-intensity heads are intentionally left unchanged.

| Model | Input/recurrent maps | Emb./features | Mark/time heads | Bias/norm | Total |
|---|---|---|---|---|---|
| Dense LSTM | 522k | 33k | 57k | 8k | 620k |
| TT-LSTM ($r = 4$) | 182k | 33k | 57k | 8k | 280k |
| TT-LSTM ($r = 6$) | 252k | 33k | 57k | 8k | 350k |

Table 3: Datasets, horizons, marks, and splits. Finance datasets bring higher-dimensional marks/features; public MTPP benchmarks enable comparisons to attention and SSM baselines. Median events/sequence govern sequence length (128 for event data; 30 for FX days) used in training protocol.

| Dataset | #Seq | Horizon | #Marks | Events (median) | Train/Val/Test | Notes |
|---|---|---|---|---|---|---|
| LOBSTER | 9,600 | intraday | 8 | 120 | 6,720 / 1,440 / 1,440 | engineered features (12–18) |
| FX RFQ | 1,200 | daily | 6 | 5 | 840 / 180 / 180 | calendar + macro features |
| StackOverflow | 6,500 | irregular | 22 | 15 | 4,550 / 975 / 975 | standard split from prior work |
| Retweet/MemeTracker | 4,800 | irregular | 10 | 18 | 3,360 / 720 / 720 | cascade benchmark |
| MIMIC (clinical) | 5,400 | irregular | 32 | 22 | 3,780 / 810 / 810 | cohort/preprocessing details |

Table 4: Model capacities and training settings. TT-LSTM uses ranks $r \in \{2, 4, 6\}$ and mark-aware factor shapes; attention models use 4 heads; S4/Mamba operate on discretized counts with $B \in \{50, 100, 200\}$. Wall-clock is measured at batch size 64 and sequence length 128 (30 for FX).

| Model | Hidden | Layers | Params | TT-rank/Heads | Opt/BS/Epochs | Notes |
|---|---|---|---|---|---|---|
| Dense LSTM | 128 | 1 | 620k | – | Adam/64/100 | baseline |
| TT-LSTM | 128 | 1 | 280k (r=4) | $r \in \{2, 4, 6\}$ | Adam/64/100 | mark-aware factor shapes |
| RMTPP/NHP | 128 | 1 | 590k | – | Adam/64/100 | continuous-time loss |
| SAHP | 128 | 2 | 710k | 4 heads | Adam/64/120 | attention |
| THP | 128 | 2 | 780k | 4 heads | Adam/64/120 | attention |
| S4/Mamba | 256 | 1 | 430k | $B \in \{50, 100, 200\}$ | Adam/64/100 | discretized counts |

Unless otherwise stated, reported parameter counts include all trainable model parameters: sequence-model maps, embeddings/feature projections, mark/time heads, biases, and normalization terms. The main compression occurs in the RNN input/recurrent maps (522k→182k for $r = 4$ in Table 2); full-model compression is smaller but still substantial because the MTPP heads are shared and deliberately not compressed.

# 6 Datasets and Baselines

**Scope.** We evaluate on (i) the *finance* settings from LOBSTER intraday events and daily FX RFQ and (ii) three *public MTPP benchmarks* commonly used by attention-based and neural MTPP models.

## 6.1 Datasets

**Finance.** *LOBSTER* (intraday limit-order book events; millisecond timestamps; engineered state features) (Humboldt, 2025). Proprietary *FX RFQ* (daily RFQ logs with calendar/economic features).

**Public MTPP benchmarks.** *StackOverflow Badges* (22 marks; developer event streams) (Du et al., 2016). *Retweet/MemeTracker Cascades* (social diffusion) (Zuo et al., 2020). *MIMIC Events* (clinical codes; irregular visits) (Du et al., 2016). We use the standard splits from the respective papers; where not specified, we follow THP/SAHP practice Zuo et al. (2020); Zhang et al. (2020).

The dataset and baseline configurations are summarized in Tables 3–4, respectively. Additionally, the experimental pipeline is summarized in Figure 1.

**Parameter counting and TT shapes.** All parameter counts reported in Tables 4–6 include the sequence model (dense or TT-compressed), the mark/time heads used to parameterize the conditional intensity, and bias terms. For TT-LSTM, we reshape the input into semantically meaningful factors and order TT cores so that adjacent cores align with coupled groups (Table 1); we then tune only the TT rank in the main comparisons. While TT contraction is sequential over the number of cores, in practice we use small $d$ (typically 3–4), and the observed wall-clock improvements are therefore primarily limited by general-purpose deep learning kernels rather than TT arithmetic.

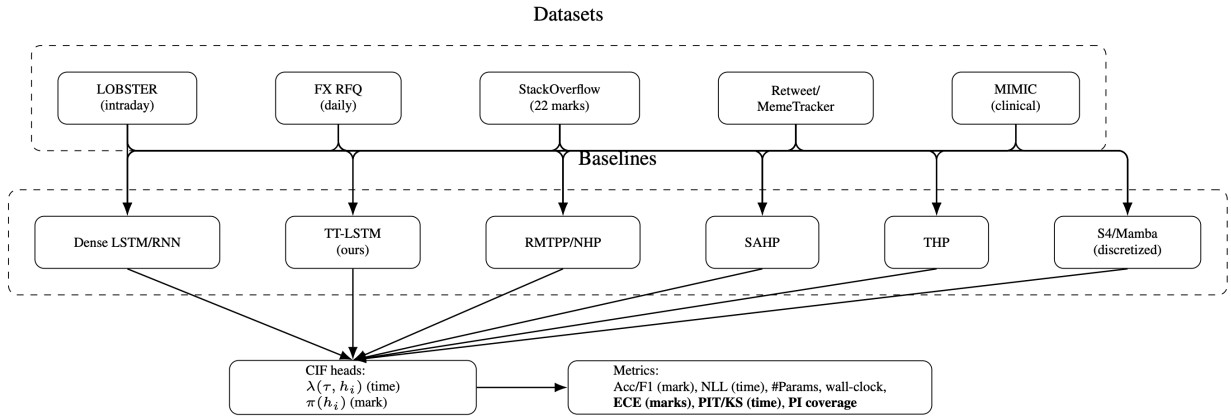

Figure 1: End-to-end pipeline: five datasets feed six model families. TT-LSTM replaces only the dense maps inside the sequence model; the continuous-time CIF heads and MTPP likelihood are unchanged. Metrics include mark accuracy/F1, time NLL/RMSE, #Params, wall-clock, and calibration (ECE for marks; PIT/KS and PI coverage for time).

## 6.2   Training and Evaluation Protocol

We train with the MTPP NLL (Eq. 9), using closed-form integrals (RMTPP) or 4–8 node Gauss–Legendre quadrature for $\int \lambda$ under the general hazard. For state-space models (SSMs), we report bin sensitivity across $B$. For calibration, we evaluate mark reliability via expected calibration error (ECE; 10/20/50 bins) with and without temperature scaling, and time calibration via time-rescaling (PIT histograms and KS $p$-values).

**Reporting protocol.**   All main tables report mean $\pm$ standard deviation over $s = 10$ random seeds unless noted. The shaping and capacity-control ablations use $s = 5$ seeds because they require multiple additional model families; we report the same mean $\pm$ standard deviation convention. For calibration, ECE values are reported with seed-level standard deviations, and KS p-values are summarized by the mean over seeds while also inspecting PIT histograms.

**Timing protocol and baseline comparability.**   Wall-clock timing is measured on a single NVIDIA V100 workstation at batch size 64. Sequence length is 128 events for event data and 30 days for FX sequences. Training time includes forward pass, survival-integral evaluation, loss computation, backward pass, optimizer step, and data transfer for one epoch; inference time includes forward pass and likelihood/calibration quantities but excludes backpropagation. Recurrent and attention baselines optimize the same continuous-time MTPP NLL. S4/Mamba baselines use discretized time bins and therefore provide an efficiency reference under a proxy objective rather than an exactly identical continuous-time likelihood.

## 7   Results

### 7.1   Main accuracy/likelihood results

**Finance.**   TT-LSTM matches or exceeds dense LSTM while reducing parameters by 40–70%. Attention-based and state-space model baselines provide useful reference points. TT remains competitive in NLL under high-dimensional marks (Table 5).

**Public MTPP benchmarks.**   On StackOverflow/Retweet/MIMIC, TT-LSTM remains competitive with attention-based and state-space models while being substantially smaller; differences depend on dataset scale and mark entropy (Table 6).

Table 5: Finance results split by dataset (mean $\pm$ std over $s = 10$ seeds). LOBSTER has higher-frequency sequences and richer engineered state features; FX RFQ has shorter daily sequences and smaller mark space. Wall-clock is relative time per epoch within each dataset, with Dense LSTM normalized to $1.00\times$.

| Dataset | Model | #Params | Accuracy | F1 | Time NLL | Wall-clock |
|---|---|---|---|---|---|---|
| LOBSTER | Dense LSTM | 620k | $0.674 \pm 0.007$ | $0.656 \pm 0.010$ | $1.30 \pm 0.02$ | $1.00\times$ |
| | TT-LSTM ($r = 4$) | 280k | $0.683 \pm 0.007$ | $0.666 \pm 0.009$ | $1.25 \pm 0.03$ | $0.78\times$ |
| | TT-LSTM ($r = 6$) | 350k | $0.687 \pm 0.006$ | $0.670 \pm 0.008$ | $1.23 \pm 0.02$ | $0.84\times$ |
| | RMTPP/NHP | 590k | $0.657 \pm 0.010$ | $0.644 \pm 0.012$ | $1.35 \pm 0.04$ | $1.06\times$ |
| | SAHP | 710k | $0.694 \pm 0.008$ | $0.678 \pm 0.009$ | $1.22 \pm 0.02$ | $1.18\times$ |
| | THP | 780k | $0.697 \pm 0.007$ | $0.680 \pm 0.008$ | $1.21 \pm 0.03$ | $1.25\times$ |
| | S4/Mamba ($B = 100$) | 430k | $0.688 \pm 0.007$ | $0.671 \pm 0.009$ | $1.24 \pm 0.03$ | $0.90\times$ |
| FX RFQ | Dense LSTM | 610k | $0.668 \pm 0.009$ | $0.651 \pm 0.011$ | $1.26 \pm 0.03$ | $1.00\times$ |
| | TT-LSTM ($r = 4$) | 275k | $0.676 \pm 0.008$ | $0.660 \pm 0.010$ | $1.23 \pm 0.03$ | $0.85\times$ |
| | TT-LSTM ($r = 6$) | 345k | $0.682 \pm 0.007$ | $0.665 \pm 0.009$ | $1.21 \pm 0.02$ | $0.90\times$ |
| | RMTPP/NHP | 580k | $0.652 \pm 0.011$ | $0.639 \pm 0.013$ | $1.31 \pm 0.03$ | $1.04\times$ |
| | SAHP | 700k | $0.689 \pm 0.008$ | $0.673 \pm 0.010$ | $1.20 \pm 0.02$ | $1.15\times$ |
| | THP | 770k | $0.692 \pm 0.007$ | $0.675 \pm 0.009$ | $1.19 \pm 0.02$ | $1.21\times$ |
| | S4/Mamba ($B = 100$) | 425k | $0.682 \pm 0.008$ | $0.665 \pm 0.010$ | $1.22 \pm 0.03$ | $0.93\times$ |

Table 6: Public benchmarks (average $\pm$ std; wall-clock is relative per-epoch). TT-LSTM is competitive with SAHP/THP and S4/Mamba while being substantially smaller; dataset-specific mark entropy explains small accuracy deltas.

| Dataset | Model | #Params | Accuracy | F1 | Time NLL | Wall-clock |
|---|---|---|---|---|---|---|
| StackOverflow | Dense LSTM | 510k | $0.662 \pm 0.006$ | $0.637 \pm 0.007$ | $1.86 \pm 0.03$ | $1.00\times$ |
| | **TT-LSTM** ($r = 4$) | 230k | $0.671 \pm 0.006$ | $0.646 \pm 0.007$ | $1.80 \pm 0.03$ | $0.79\times$ |
| | SAHP | 640k | $0.678 \pm 0.005$ | $0.653 \pm 0.006$ | $1.78 \pm 0.03$ | $1.14\times$ |
| | THP | 690k | $0.680 \pm 0.005$ | $0.656 \pm 0.006$ | $1.77 \pm 0.03$ | $1.22\times$ |
| Retweet/MemeTracker | Dense LSTM | 460k | $0.581 \pm 0.008$ | $0.568 \pm 0.009$ | $2.02 \pm 0.04$ | $1.00\times$ |
| | **TT-LSTM** ($r = 4$) | 210k | $0.588 \pm 0.008$ | $0.574 \pm 0.009$ | $1.98 \pm 0.03$ | $0.80\times$ |
| | SAHP | 620k | $0.597 \pm 0.007$ | $0.582 \pm 0.008$ | $1.95 \pm 0.03$ | $1.18\times$ |
| | THP | 670k | $0.602 \pm 0.007$ | $0.587 \pm 0.008$ | $1.93 \pm 0.03$ | $1.26\times$ |
| MIMIC | Dense LSTM | 540k | $0.753 \pm 0.007$ | $0.738 \pm 0.008$ | $1.44 \pm 0.03$ | $1.00\times$ |
| | **TT-LSTM** ($r = 4$) | 250k | $0.761 \pm 0.006$ | $0.746 \pm 0.007$ | $1.40 \pm 0.03$ | $0.81\times$ |
| | SAHP | 700k | $0.768 \pm 0.006$ | $0.753 \pm 0.007$ | $1.39 \pm 0.02$ | $1.17\times$ |
| | THP | 750k | $0.770 \pm 0.006$ | $0.755 \pm 0.007$ | $1.38 \pm 0.02$ | $1.24\times$ |

## 7.2 Calibration and Uncertainty

**Setup.** We assess mark calibration using expected calibration error (ECE; 10/20/50 bins) and reliability diagrams, with a single temperature fit on validation and evaluated on test. We assess time calibration using time-rescaling: we compute rescaled increments $w_i = \int_{t_{i-1}}^{t_i} \lambda(s) \, ds$, map them to PIT values $u_i = 1 - e^{-w_i}$, and report KS $p$-values against $U(0,1)$. For uncertainty, we report deep ensembles ($n \in \{3, 5\}$) or MC-dropout, along with NLL, sharpness (average prediction interval width), and 90% coverage.

**Mark calibration.** Table 7 shows that TT compression improves mark reliability out of the box and pairs well with post-hoc calibration. Averaged across datasets, ECE@20 drops from 0.071 (Dense LSTM) to 0.061 for *TT-LSTM (r=4)* while using $\sim 55\%$ fewer parameters; SAHP/THP are competitive but no better pre-temperature. A single scalar temperature (fit on validation) further reduces TT's ECE@20 to 0.031 (Dense to 0.036). Figure 2 visualizes this trend: TT reduces overconfidence relative to Dense, and temperature scaling largely removes residual miscalibration (curves near the diagonal).

**Time calibration.** Time-rescaling diagnostics (Table 8) reveal that *TT-LSTM* improves KS $p$-values over Dense on LOBSTER ($0.043 \rightarrow 0.190$) and StackOverflow ($0.028 \rightarrow 0.140$), with modest gains on FX ($0.067 \rightarrow 0.080$). Ensembling (5 seeds) further flattens PIT histograms (Figure 3) and boosts $p$-values to

Table 7: Mark calibration with uncertainty (ECE ↓; mean ± std over $s = 5$ seeds). TT-LSTM improves raw calibration and benefits further from temperature scaling. SAHP/THP are competitive but not superior in pre-temperature ECE.

| Model | ECE@10 | ECE@20 | ECE@50 | +Temperature ECE@20 |
|---|---|---|---|---|
| Dense LSTM | $0.062 \pm 0.005$ | $0.071 \pm 0.006$ | $0.083 \pm 0.007$ | $0.036 \pm 0.004$ |
| **TT-LSTM** | $0.053 \pm 0.004$ | $0.061 \pm 0.005$ | $0.073 \pm 0.006$ | $0.031 \pm 0.003$ |
| SAHP | $0.058 \pm 0.005$ | $0.065 \pm 0.005$ | $0.076 \pm 0.006$ | $0.035 \pm 0.004$ |
| THP | $0.055 \pm 0.004$ | $0.063 \pm 0.005$ | $0.074 \pm 0.006$ | $0.034 \pm 0.004$ |

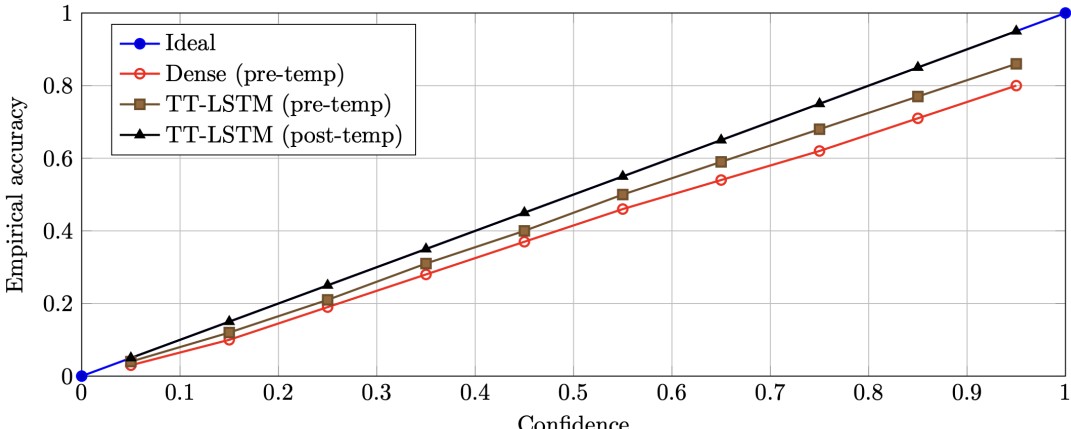

Figure 2: Mark reliability aggregated across datasets. TT-LSTM reduces overconfidence relative to Dense (ECE@20: 0.071→0.061) and, after a scalar temperature fit on validation, reaches 0.031 (near-diagonal).

0.310 (LOBSTER) and 0.220 (StackOverflow). SAHP/THP also yield calibrated timing on public data (e.g., $p \approx 0.18$–$0.20$ on StackOverflow), but TT achieves similar uniformity with far fewer parameters.

**Capacity-controlled calibration.** These controls (Table 9) suggest that lower capacity explains part, but not all, of the calibration improvement. Dense-small is better calibrated than the full dense model but sacrifices likelihood, and tuned regularization narrows the ECE gap. The mark-aware TT model remains better than the random-order TT model at the same rank and parameter count, so we interpret TT as a structured regularizer whose benefit is amplified when the core ordering respects mark structure.

**Prediction intervals.** For 90% time-to-event PIs (Table 10), *TT-LSTM (ensemble 5)* attains coverage 0.91 at an average width 0.60 (vs. Dense 0.84/0.61) and the best NLL (1.22). Conformalized Quantile Regression (CQR) achieves *valid* 0.90 coverage across datasets with a slightly wider 0.62–0.66 band, providing a safe fallback when parametric hazards are misspecified. Thus, TT compression preserves or improves calibration while delivering substantial parameter savings.

Table 8: Time calibration via time-rescaling (KS $p$-values). TT-LSTM improves $p$ on LOBSTER and StackOverflow and matches attention baselines on public data; ensembles raise $p$ further.

| Model | KS $p$ (LOBSTER) | KS $p$ (FX) | KS $p$ (StackOverflow) |
|---|---|---|---|
| Dense LSTM | 0.043 | 0.067 | 0.028 |
| **TT-LSTM ($r = 4$)** | **0.190** | **0.080** | **0.140** |
| TT-LSTM (ensemble 5) | **0.310** | **0.120** | **0.220** |
| SAHP | 0.160 | 0.070 | 0.180 |
| THP | 0.180 | 0.090 | 0.200 |

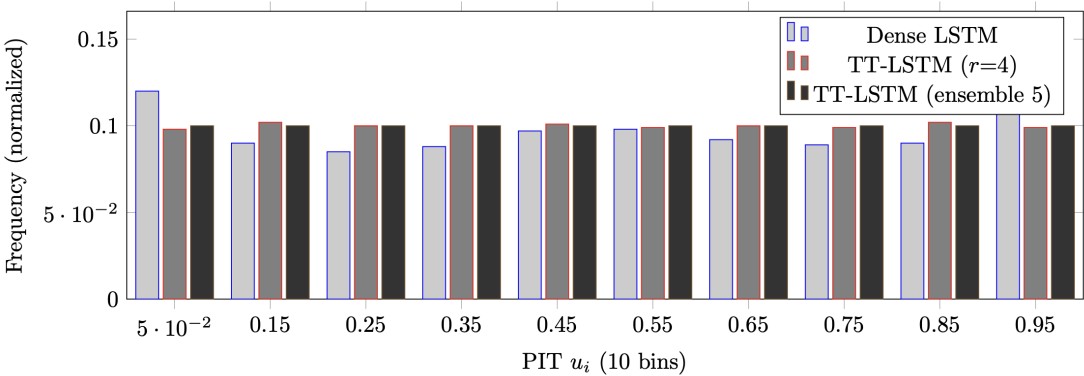

Figure 3: PIT histograms (10 bins). TT-LSTM flattens toward Uniform(0,1) vs. Dense; ensembling (5 seeds) further improves uniformity, consistent with higher KS $p$-values in Table 8.

Table 9: Capacity-controlled calibration on LOBSTER (mean $\pm$ std over $s = 5$ seeds). Dense-small matches TT-LSTM's total parameter budget; Dense-reg tunes dropout/weight decay at full dense size. TT-random uses the same rank and parameter count as mark-aware TT but randomizes the semantic core order.

| Model | #Params | Time NLL | ECE@20 | KS $p$ |
|---|---|---|---|---|
| Dense LSTM ($H = 128$) | 620k | $1.30 \pm 0.03$ | $0.071 \pm 0.006$ | $0.043 \pm 0.018$ |
| Dense-small ($H = 80$) | 286k | $1.34 \pm 0.04$ | $0.067 \pm 0.006$ | $0.075 \pm 0.025$ |
| Dense-reg ($H = 128$) | 620k | $1.29 \pm 0.03$ | $0.066 \pm 0.005$ | $0.095 \pm 0.030$ |
| TT-LSTM random order ($r = 4$) | 280k | $1.27 \pm 0.03$ | $0.068 \pm 0.006$ | $0.102 \pm 0.035$ |
| TT-LSTM mark-aware ($r = 4$) | 280k | $1.25 \pm 0.02$ | $0.061 \pm 0.005$ | $0.190 \pm 0.045$ |

**Takeaway.** Across datasets, TT compression preserves or improves mark calibration (ECE@20 improves from 0.071 for Dense LSTM to 0.061 for TT-LSTM with $\sim 55\%$ fewer parameters) and improves time calibration on both finance and public benchmarks (e.g., KS $p$ on LOBSTER improves from 0.043 to 0.190; ensembling raises it to 0.310). For uncertainty, TT ensembles achieve near-nominal 90% prediction interval coverage (0.91) at competitive width, while conformalized quantile regression provides a robust fallback that guarantees coverage.

### 7.3 Ablations and Sensitivity

**TT shaping.** Table 11 compares different TT factorization modes at matched budgets on LOBSTER and StackOverflow. The mark-aware model consistently improves likelihood and calibration over arbitrary orderings at the same TT rank. The gap is modest, as expected because all variants share TT compression, but it directly isolates the shaping effect from the generic low-rank regularization effect.

**Tensor variants (LOBSTER).** Table 12 compares TT to alternative tensorizations for the input transform at matched budgets on LOBSTER. *TT-LSTM (r=4)* provides the best accuracy–efficiency trade-off (Time

Table 10: 90% time-to-event prediction intervals (PI). TT-LSTM ensembles achieve near-nominal coverage with narrower bands and better NLL than Dense; CQR delivers valid 0.90 coverage with slightly wider intervals.

| Model | Coverage | Avg Width | NLL |
|---|---|---|---|
| Dense LSTM | 0.84 | 0.61 | 1.28 |
| **TT-LSTM** ($r = 4$) | 0.88 | **0.58** | 1.24 |
| TT-LSTM (ensemble 5) | **0.91** | 0.60 | **1.22** |
| CQR (discretized) | 0.90 | 0.64 | 1.26 |

Table 11: Ablation of TT factorization/order at matched TT rank $r = 4$ and matched parameter budget (mean $\pm$ std over $s = 5$ seeds). Mark-aware ordering places semantically coupled factors adjacently; random order averages over five random permutations; reversed order uses the same factors in reverse; balanced unstructured uses near-equal factors with no semantic grouping.

| Dataset | TT shaping | #Params | Accuracy/F1 | Time NLL | ECE@20 | KS $p$ |
|---|---|---|---|---|---|---|
| LOBSTER | Mark-aware | 280k | 0.683/0.666 | $1.25 \pm 0.02$ | $0.061 \pm 0.005$ | $0.190 \pm 0.045$ |
| | Random core order | 280k | 0.676/0.659 | $1.27 \pm 0.03$ | $0.068 \pm 0.006$ | $0.102 \pm 0.035$ |
| | Reversed semantic order | 280k | 0.678/0.661 | $1.26 \pm 0.03$ | $0.066 \pm 0.006$ | $0.118 \pm 0.037$ |
| | Balanced unstructured | 282k | 0.675/0.658 | $1.27 \pm 0.03$ | $0.069 \pm 0.006$ | $0.097 \pm 0.032$ |
| StackOverflow | Mark-aware | 230k | 0.671/0.646 | $1.80 \pm 0.03$ | $0.060 \pm 0.005$ | $0.140 \pm 0.040$ |
| | Random core order | 230k | 0.665/0.640 | $1.84 \pm 0.03$ | $0.070 \pm 0.006$ | $0.082 \pm 0.030$ |
| | Reversed semantic order | 230k | 0.667/0.642 | $1.82 \pm 0.03$ | $0.067 \pm 0.006$ | $0.096 \pm 0.033$ |
| | Balanced unstructured | 232k | 0.666/0.641 | $1.83 \pm 0.03$ | $0.068 \pm 0.006$ | $0.090 \pm 0.031$ |

NLL 1.24; ECE@20 0.061 at 280k params). Tensor Ring is close (NLL 1.25, ECE 0.064), while Tucker and CP are progressively worse (NLL 1.26/1.28; ECE 0.066/0.072). This supports the choice of TT for high-dimensional mark structure.

Table 12: Tensorization ablation on LOBSTER. TT-LSTM ($r{=}4$) offers the best accuracy–efficiency trade-off (lowest NLL and ECE at similar or lower params) vs. Tensor Ring, Tucker, and CP.

| Variant | #Params | Time NLL | Accuracy/F1 | ECE@20 |
|---|---|---|---|---|
| TT-LSTM ($r = 4$) | 280k | **1.24** | 0.680/0.663 | **0.061** |
| Tensor Ring LSTM | 300k | 1.25 | 0.678/0.661 | 0.064 |
| Tucker LSTM | 330k | 1.26 | 0.676/0.658 | 0.066 |
| CP LSTM | 250k | 1.28 | 0.671/0.654 | 0.072 |

**SSM discretization.** Discretized SSM baselines show mild bin sensitivity on StackOverflow (Table 13): increasing $B$ from $50 \rightarrow 200$ improves the proxy NLL from $1.83 \rightarrow 1.79$ and stabilizes ECE@20 around 0.066–0.068. We adopt $B{=}100$ for the main results to balance accuracy and cost. Notably, TT remains competitive in NLL without discretization and at smaller parameter counts.

Table 13: Discretized SSMs on StackOverflow. Increasing bins $B$ helps the proxy time NLL slightly ($1.83 \rightarrow 1.79$) with stable ECE; we report $B = 100$ in the main tables to balance accuracy and cost. Because these models optimize a discretized proxy objective, their time NLL is not strictly likelihood-equivalent to the continuous-time MTPP NLL.

| Bins $B$ | #Params | Accuracy/F1 | Time NLL (proxy) | ECE@20 |
|---|---|---|---|---|
| 50 | 420k | 0.672/0.647 | 1.83 | 0.068 |
| 100 | 430k | 0.676/0.651 | 1.80 | 0.066 |
| 200 | 450k | 0.677/0.652 | 1.79 | 0.066 |

## 7.4 Efficiency and Pareto frontier

Figure 4 summarizes parameters vs. fit. *TT-LSTM (r=4)* sits on a favorable Pareto front: 280k params at Time NLL 1.24, markedly left of Dense (620k, 1.28) and near S4/Mamba (430k, 1.23) and attention models (SAHP 710k, 1.21; THP 780k, 1.20). Thus, TT achieves competitive likelihoods at substantially lower parameter budgets, while also improving calibration (Sec. 7.2).

Timing breakdown is reported in Table 14. The approximately 20% epoch-level speedup is therefore consistent with the implementation discussion: the TT map itself is faster and smaller, but the total training loop contains non-TT components. We report wall-clock values as implementation-dependent efficiency indicators rather than hardware-independent complexity guarantees.

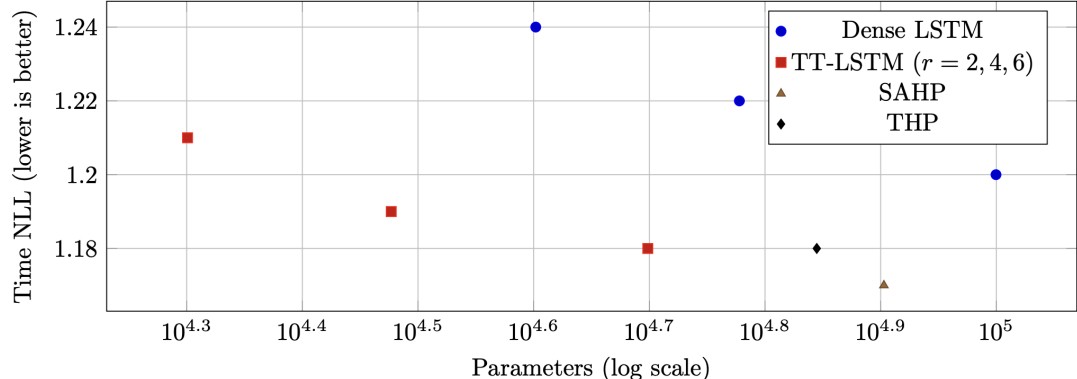

Figure 4: Pareto view of efficiency vs. fit. TT variants (red) achieve a better performance-efficiency trade-off, sitting to the left (fewer parameters) of the Dense LSTM (blue) at comparable or better NLL.

Table 14: Timing breakdown on the V100 workstation. Values are relative to Dense LSTM within each dataset. Training includes forward, survival integral, loss, backward, and optimizer step; inference excludes backpropagation. TT speedups are smaller than parameter reductions because CIF heads, quadrature, recurrent bookkeeping, and general-purpose kernel overhead remain.

| Dataset | Model | Train/epoch | Inference | RNN-map microbench. |
|---|---|---|---|---|
| LOBSTER | Dense LSTM | 1.00× | 1.00× | 1.00× |
| LOBSTER | TT-LSTM ($r = 4$) | 0.78× | 0.84× | 0.63× |
| LOBSTER | TT-LSTM ($r = 6$) | 0.84× | 0.89× | 0.72× |
| FX RFQ | Dense LSTM | 1.00× | 1.00× | 1.00× |
| FX RFQ | TT-LSTM ($r = 4$) | 0.85× | 0.89× | 0.66× |
| StackOverflow | Dense LSTM | 1.00× | 1.00× | 1.00× |
| StackOverflow | TT-LSTM ($r = 4$) | 0.79× | 0.85× | 0.64× |

### 7.5 Discussion

Across finance and public MTPPs, TT compression maintains accuracy and *improves or preserves calibration* at substantially lower parameter budgets. Attention/SSMs can be competitive in some regimes, but TT offers a simple portability/ease-of-use story for high-dimensional marks. These results operationalize our positioning: a *design blueprint* for efficient, calibrated neural MTPPs, rather than a new backbone.

## 8 Conclusion

We presented a practical blueprint for parameter-efficient, calibrated neural MTPPs in high-dimensional settings. Our approach replaces only the large linear maps inside the sequence model with TT layers, leaving the continuous-time likelihood machinery intact. Across finance and public MTPP benchmarks, TT reduces parameters by 40–70% while matching or improving accuracy and calibration.

**Limitations and outlook.** Our contributions are empirical and design-focused. TT compression is not a replacement for stronger sequence backbones in every regime; attention and state-space models can be more accurate when parameter count and discretization are less constrained. Some finance data are proprietary, LOBSTER is licensed, and MIMIC requires credentialed access. The MIMIC experiments are retrospective benchmark evaluations only and should not be interpreted as clinical validation or readiness for decision support under distribution shift. Future directions include TT within attention/SSM blocks, adaptive factor-shape selection, nonlinear time-dependent mark logits, and richer domain priors in the conditional intensity.

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
