## Appendix

## Appendix Table of Contents

## A  Additional Methodological Details

### A.1  CIF, likelihood, and quadrature details

Let $\mathcal{D} = \{(t_i, x_i)\}_{i=1}^N$ with history $\mathcal{H}_t$. The marked intensity $\lambda_k(t \mid \mathcal{H}_t)$ and total $\lambda(t \mid \mathcal{H}_t) = \sum_k \lambda_k(\cdot)$ define the joint next-event density $f(t_{i+1}, x_{i+1} | \mathcal{H}_{t_i}) = \lambda_{x_{i+1}}(t_{i+1} | \mathcal{H}_{t_{i+1}}) \exp(-\int_{t_i}^{t_{i+1}} \lambda)$. The negative log-likelihood on $[0, T]$ is $\mathcal{L}_{\text{TPP}} = -\sum_{i=1}^N \log \lambda_{x_i}(t_i \mid \mathcal{H}_{t_i}) + \int_0^T \lambda(s \mid \mathcal{H}_s) \, ds$. Under the decomposition $\lambda_k = \lambda \cdot \pi_k$, this splits into time and mark terms (main text Eq. 10).

**Quadrature.**   For the general neural hazard (main text Eq. 8), we evaluate $\int_{t_i}^{t_{i+1}} \lambda$ by Gauss–Legendre with $n_q \in \{4, 8\}$ nodes; the per-interval error is $O(n_q^{-2})$. We cache $\phi_\tau(\tau)$ to reduce overhead. For RMTPP, the integral has a closed form.

### A.2  Time-rescaling theorem and PIT/KS diagnostics

Let $w_i = \int_{t_{i-1}}^{t_i} \lambda(s \mid \mathcal{H}_s) \, ds$ and $u_i = 1 - e^{-w_i}$. Under correct specification, $\{w_i\}$ are i.i.d. Exp(1), hence $\{u_i\}$ are i.i.d. Uniform$(0, 1)$ (Brown et al., 2002; Ogata, 1988). We evaluate goodness-of-fit with PIT histograms and a one-sample KS test. See Algorithm 2 for the recipe.

---

**Algorithm 2** Time-rescaling and PIT/KS diagnostics

**Require:** Trained model parameters $\theta$, test sequence $\{(t_i, x_i)\}_{i=1}^N$
1: Compute hidden states $\{h_i\}$ via Eq. 6.
2: **for** $i = 2$ to $N$ **do**
3:    Set $\tau \leftarrow t - t_{i-1}$; define $\lambda(\tau, h_{i-1})$ per Eq. 8 or RMTPP form.
4:    Evaluate $w_i = \int_{t_{i-1}}^{t_i} \lambda(s \mid \mathcal{H}_s) \, ds$ (closed form or Gauss–Legendre).
5:    Set $u_i = 1 - \exp(-w_i)$.
6: **end for**
7: Perform one-sample KS test of $\{u_i\}$ against $U(0, 1)$; plot PIT histogram and Q–Q.
8: Return: KS $p$-value, histogram, Q–Q plot.

---

### A.3 TT Layer Algorithm

**Algorithm 3** represents an RNN cell (e.g., a simple LSTM) using TT layers internally. This supplements the discussion in Section 5.2 in the main text.

---

**Algorithm 3** TT-based RNN Cell (e.g., LSTM variant, schematic)

---

1: **Input:** current input $\mathbf{x}_t$, previous state $(\mathbf{h}_{t-1}, \mathbf{c}_{t-1})$
2: **Output:** new hidden state $\mathbf{h}_t, \mathbf{c}_t$

3: $\mathbf{z}_{\text{in}} \leftarrow \text{TTLayer}(\mathbf{x}_t)$   (input transform)
4: $\mathbf{z}_{\text{rec}} \leftarrow \text{TTLayer}(\mathbf{h}_{t-1})$   (recurrent transform)
5: $\mathbf{z} \leftarrow \mathbf{z}_{\text{in}} + \mathbf{z}_{\text{rec}}$
6: Split $\mathbf{z}$ into $(\mathbf{z}^i, \mathbf{z}^f, \mathbf{z}^o, \mathbf{z}^g)$ for gates
7: $\mathbf{i}_t \leftarrow \sigma(\mathbf{z}^i)$   (input gate)
8: $\mathbf{f}_t \leftarrow \sigma(\mathbf{z}^f)$   (forget gate)
9: $\mathbf{o}_t \leftarrow \sigma(\mathbf{z}^o)$   (output gate)
10: $\mathbf{g}_t \leftarrow \tanh(\mathbf{z}^g)$   (candidate)
11: $\mathbf{c}_t \leftarrow \mathbf{f}_t \odot \mathbf{c}_{t-1} + \mathbf{i}_t \odot \mathbf{g}_t$
12: $\mathbf{h}_t \leftarrow \mathbf{o}_t \odot \tanh(\mathbf{c}_t)$
13: **return** $\mathbf{h}_t, \mathbf{c}_t$

---

### A.4 Mark-aware TT shaping

Table 1 in the main text shows how TT differs from a generic low-rank layer and supports our "problem-driven design" position discussed in Section 5.2 in the main text.

**Dense vs. TT-Layer.** A standard dense $Wx + b$ operation inside an RNN cell (right) compared to the TT-Layer operation (left) is depicted in Figure 5. The input tensor $x$ is reshaped according to the mark-aware factors $(m_1, m_2, \dots)$ before being processed by the TT-cores.

## B  Additional Experiments

### B.1  Conformalized Quantile Regression for Time-to-Event PIs

**Goal.** Construct marginally valid $(1 - \alpha)$ prediction intervals (PIs) for inter-event times $\Delta t_{i+1} = t_{i+1} - t_i$ given history $\mathcal{H}_{t_i}$, with no distributional assumptions beyond exchangeability on the calibration set (Romano et al., 2019).

**Base quantile model.** We train two quantile regressors $\hat{q}_\ell(\cdot), \hat{q}_u(\cdot)$ for levels $\tau_\ell = \alpha/2$ and $\tau_u = 1 - \alpha/2$ using pinball losses. Inputs are the hidden state and time features: $x_i = [h_i; \phi_\tau(0)]$ (or optionally $[h_i; \phi_\tau(\tau)]$ with $\tau{=}0$ as a marker). We use the same backbone (Dense or TT-LSTM) as in the main model, but the heads output quantiles for $\Delta t_{i+1}$.

**Conformalization.** On a held-out calibration set $\mathcal{C}$, compute nonconformity scores $s_j = \max\big\{\hat{q}_\ell(x_j) - y_j, \ y_j - \hat{q}_u(x_j), \ 0\big\}$. Let $q_{1-\alpha}$ be the $\lceil(1-\alpha)(|\mathcal{C}| + 1)\rceil$-th order statistic of $\{s_j\}_{j \in \mathcal{C}}$. At test time, the conformal PI is

$$\big[\, \hat{q}_\ell(x) - q_{1-\alpha}, \ \hat{q}_u(x) + q_{1-\alpha} \,\big]$$

which achieves finite-sample marginal coverage $\geq 1 - \alpha$ under exchangeability.

**Variants.** (i) *Direct CQR* on $\Delta t$ as above. (ii) *Residual conformalization* around an RMTPP-style parametric quantile (closed-form from $\lambda$) to obtain tighter PIs when the parametric form is well-specified.

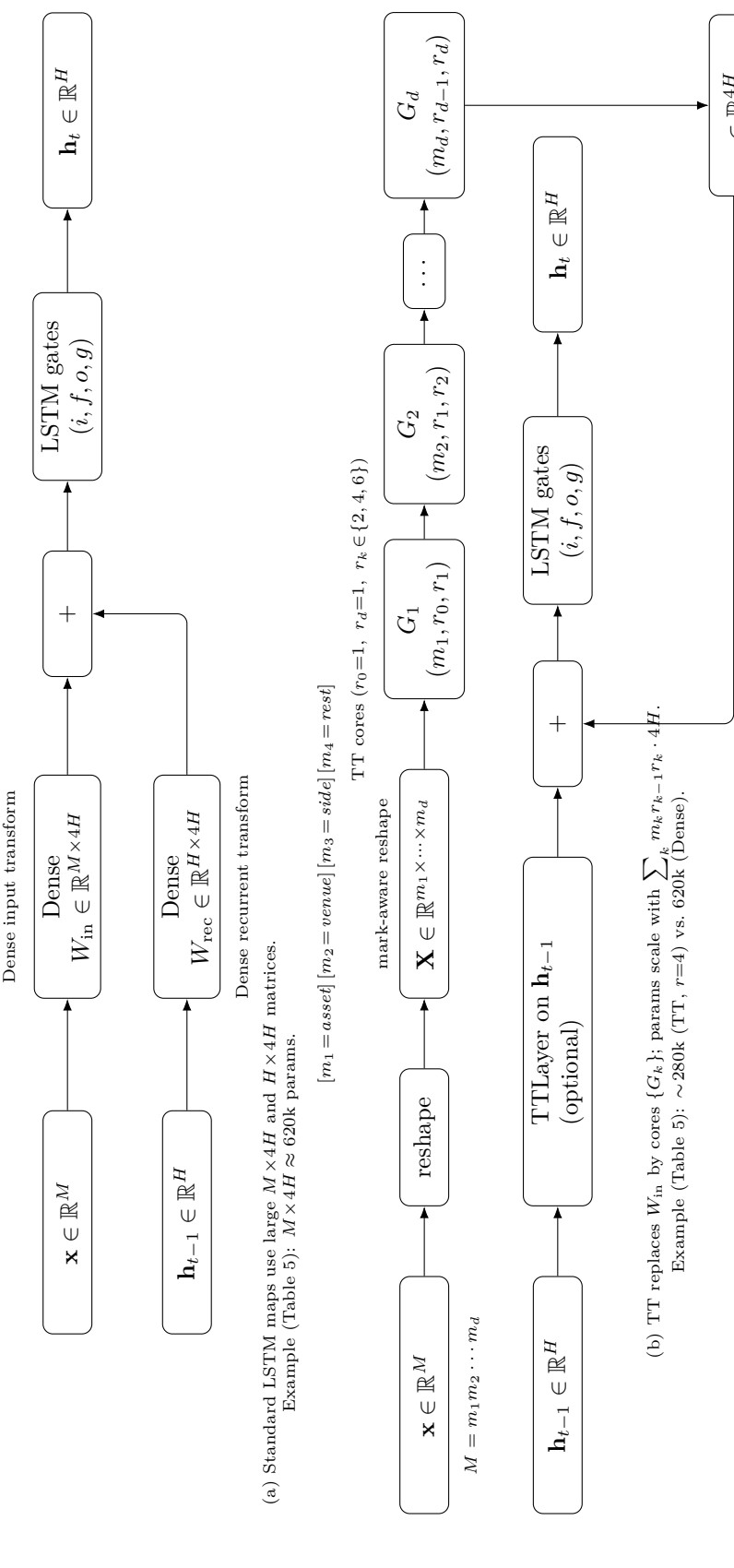

(a) Standard LSTM maps use large $M \times 4H$ and $H \times 4H$ matrices.
Example (Table 5): $M \times 4H \approx 620k$ params.

$[m_1 = asset]\,[m_2 = venue]\,[m_3 = side]\,[m_4 = rest]$

TT cores ($r_0 = 1$, $r_d = 1$, $r_k \in \{2, 4, 6\}$)

mark-aware reshape

$M = m_1 m_2 \cdots m_d$

(b) TT replaces $W_{\text{in}}$ by cores $\{G_k\}$; params scale with $\sum_k m_k r_{k-1} r_k \cdot 4H$.
Example (Table 5): $\sim 280k$ (TT, $r{=}4$) vs. 620k (Dense).

Figure 5: Dense vs. TT input transforms inside an LSTM cell with *mark-aware* shaping. (a) Dense uses large $M \times 4H$ and $H \times 4H$ matrices. (b) TT reshapes $\mathbf{x}$ into $m_1 \times \cdots \times m_d$ factors aligned to domain groups (e.g., asset, venue, side), then contracts via TT cores to produce the $4H$ gate vector. This preserves the CIF/loss head while cutting parameters (e.g., 620k $\rightarrow$ 280k for $r{=}4$) and improves calibration in Sec. 7.2.

Table 15: CQR vs. model-based PIs (per dataset). Target coverage $= 0.90$.

| Dataset | Method | Coverage | Avg Width | Notes |
|---|---|---|---|---|
| LOBSTER | TT-LSTM ($r=4$) | 0.89 | 0.56 | model PI |
| | TT-LSTM (ens. 5) | **0.91** | 0.58 | model PI |
| | **CQR (direct)** | 0.90 | 0.62 | valid by design |
| FX RFQ | TT-LSTM ($r=4$) | 0.88 | **0.60** | model PI |
| | TT-LSTM (ens. 5) | **0.90** | 0.62 | model PI |
| | **CQR (direct)** | 0.90 | 0.66 | valid by design |
| StackOverflow | TT-LSTM ($r=4$) | 0.87 | 0.59 | model PI |
| | TT-LSTM (ens. 5) | **0.90** | 0.61 | model PI |
| | **CQR (direct)** | 0.90 | 0.64 | valid by design |

**Algorithm.** See Alg. 4.

---

**Algorithm 4** Conformalized Quantile Regression for $\Delta t$

---

**Require:** Train set $\mathcal{T}$, calibration set $\mathcal{C}$, levels $(\tau_\ell, \tau_u)$, target coverage $1 - \alpha$
  1: Train $\hat{q}_\ell, \hat{q}_u$ on $\mathcal{T}$ via pinball losses at $(\tau_\ell, \tau_u)$ using inputs $x_i = [h_i; \phi_\tau(0)]$, targets $y_i = \Delta t_{i+1}$.
  2: For each $(x_j, y_j) \in \mathcal{C}$ compute $s_j = \max\{\hat{q}_\ell(x_j) - y_j, \ y_j - \hat{q}_u(x_j), \ 0\}$.
  3: Let $q_{1-\alpha} = \text{Quantile}_{\lceil (1-\alpha)(|\mathcal{C}|+1) \rceil}(\{s_j\})$.
  4: At test time, output PI $[\hat{q}_\ell(x) - q_{1-\alpha}, \ \hat{q}_u(x) + q_{1-\alpha}]$.

---

**Results (90% PIs; averages across seeds).** Table 15 details dataset-wise coverage and sharpness, consistent with main-text Table 10.

**Takeaways.** CQR attains nominal coverage with slightly wider intervals than model-based PIs; ensembles reduce miscalibration in the model-based approach and close the gap in both coverage and NLL. We view CQR as a reliable fallback when hazard parameterization is misspecified or binning is required.

## C    Additional Implementation Details

### C.1    Dataset preprocessing

**LOBSTER.** We use licensed LOBSTER message/order-book data and construct event sequences by aggregating millisecond messages into 1–2 second windows. Marks are derived from event type/side combinations; exogenous features include event counts, submitted/cancelled volume, spread, mid-price change, order-book imbalance, and selected depth statistics. Continuous features are standardized using train-set statistics only. Splits preserve chronological order to avoid leakage.

**FX RFQ.** The proprietary FX RFQ data are aggregated at daily granularity. Marks correspond to RFQ category/pair groups; exogenous features include calendar flags, day-of-week/month indicators, and macro-announcement indicators. Non-binary covariates are standardized using training statistics. The dataset cannot be redistributed because of contractual restrictions.

**StackOverflow.** We follow the standard StackOverflow badge event benchmark: each user sequence contains badge events with 22 mark types. We preserve the standard train/validation/test split when available; otherwise, we split by sequence while preserving within-sequence time order. Times are normalized per sequence by the training-set scale parameter, and rare/empty sequences are filtered following prior neural MTPP practice.

**Retweet/MemeTracker.** We use cascade sequences from the Retweet/MemeTracker benchmark. Each cascade is treated as one sequence; marks correspond to event/source categories used in prior THP/SAHP

evaluations. We remove cascades shorter than the minimum length required for next-event prediction, normalize times using train-set statistics, and split by cascade to avoid sharing events across splits.

**MIMIC.** We use retrospective clinical event sequences with diagnosis/procedure/code groups as marks. We map raw codes to a reduced vocabulary using clinically meaningful groupings, filter extremely rare marks, and normalize irregular visit times. MIMIC access requires credentialing and a data-use agreement; our release will include preprocessing scripts but not raw clinical data.

## C.2 Training & Inference

**Hardware.** All runs were on a single workstation:

- CPU: $2\times$ Intel Xeon (20 cores each)

- GPU: $1\times$ NVIDIA V100

- RAM: 64 GB

**Software.**

- Python 3.10, PyTorch

- NumPy, SciPy, scikit-learn (metrics; KS test; quantile loss)

**Optimization & schedules.** Unless stated otherwise, we use Adam (betas $= (0.9, 0.999)$, weight decay $= 10^{-4}$), initial learning rate $1\times10^{-3}$, gradient clipping at 1.0, and early stopping (patience $= 10$ epochs). Batch size is **64**. Sequence length is **128** events (LOBSTER, StackOverflow, Retweet, MIMIC) and **30** days (FX).

**Likelihood, quadrature, and calibration.** We train with the MTPP NLL (Secs. 3–4) using closed form for RMTPP and $n_q \in \{4, 8\}$ Gauss–Legendre nodes for general hazards. For mark calibration we report ECE with 10/20/50 bins and fit scalar temperature on the validation set. Time calibration uses the time-rescaling theorem (PIT histograms, one-sample KS). Ensembles average $n = 5$ independently seeded models at test time. Conformalized Quantile Regression (CQR) follows Appendix B.1.

**S4/Mamba setup.** We discretize time into $B \in \{50, 100, 200\}$ bins and train on bin-wise counts; we report bin sensitivity in Table 13.

**Mark-aware TT shaping.** Input features/marks are reshaped into semantically meaningful factors (e.g., $asset \times venue \times side \times rest$ for LOBSTER; $family \times subtype \times rest$ for StackOverflow; see Table 1). TT ranks $r \in \{2, 4, 6\}$; headline results use $r = 4$. This preserves the CIF head and loss unchanged while reducing parameters (e.g., 620k$\rightarrow$ 280k).

**Choosing TT factor shapes.** When semantic groups are available, we choose input factors to match those groups and place strongly coupled groups adjacently in the TT chain (e.g., asset–venue–side–rest for LOBSTER; family–subtype–rest for StackOverflow). When no semantic grouping is available, we use a balanced factorization obtained from the prime factors of the input/output dimensions, minimize padding, and select among a small number of core orderings on validation NLL/ECE. We recommend tuning TT rank first and core ordering second; in our experiments, rank has the largest effect on likelihood, while semantic ordering mainly improves calibration and small-NLL differences.

## C.3 Hyperparameters

We report hyperparameters in Table 16.

**Fixed defaults (for reproducibility).**

- Weight decay $= 10^{-4}$; gradient clip $= 1.0$; max epochs $= 100$ (SAHP/THP: 120); seed set $\{0, 1, 2, 3, 4\}$ (ensembles use seeds $\{0, 1, 2, 3, 4\}$).

- ECE computed as weighted bin gap (bins of equal width). KS is a one-sample test on PIT values.

- Train/val/test splits match Table 3; no sequence leakage (time order preserved).

### C.4 Reproducibility Statement

**Data access.**  LOBSTER/MIMIC are subscription-based; StackOverflow/Retweet are public (see Sec. 6, Table 3). We provide *all hyper-parameters*, the full training protocol, factor shapes (Table 1), and evaluation recipes (ECE, PIT/KS, CQR). We will release a reference implementation of TT-LSTM layers and the calibration/evaluation pipeline, together with scripts to reproduce results on the public benchmarks. Finance and clinical datasets (LOBSTER; proprietary FX RFQ; MIMIC) cannot be redistributed.

**Splits and seeds.**  Splits follow Table 3. Seeds for single-model results are $\{0, \dots, 9\}$ (report average $\pm$ std); ensembles use seeds $\{0, \dots, 4\}$.

**Implementation notes.**  Batch size 64; sequence length 128 events (30 days for FX). Adam ($10^{-3}$), weight decay $10^{-4}$, gradient clip 1.0, early stopping (patience $= 10$), quadrature nodes $n_q \in \{4, 8\}$. Temperature scaling uses LBFGS (10 steps) on validation NLL; PIT uses Gauss–Legendre for $\int \lambda$; CQR follows Appendix B.1.

**Complexity.**  Dense input map: $\mathcal{O}(M \cdot 4H)$ parameters and time; TT map: $\mathcal{O}\big(\sum_{k=1}^{d} m_k r_{k-1} r_k \cdot 4H\big)$ parameters and proportional compute.

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

Table 16: Key settings, search grids, and final selections (per model family). All models trained with Adam, LR $1\times10^{-3}$ unless noted, batch size 64, sequence length 128 (events) / 30 (FX days), early stopping (patience = 10).

| Component | Grid / Setting | Selected (headline) | Rationale |
|---|---|---|---|
| Dense LSTM | $H \in \{64, 128, 192\}$; Layers $\in \{1, 2\}$; Dropout $\in \{0, 0.1, 0.2\}$ | $H = 128$, Layers $= 1$, Dropout $= 0.1$ | Strong baseline w/ moderate capacity |
| TT-LSTM (ours) | $r \in \{2, 4, 6\}$; $d \in \{3, 4\}$; mark-aware factor shapes; Dropout $\in \{0, 0.1\}$ | $r = 4$, $d = 4$, Dropout $= 0.1$ | Pareto best (Table 5) |
| RMTPP/NHP | $H \in \{64, 128\}$; time–head: RMTPP (closed form) vs. general hazard (quad) | $H = 128$, RMTPP head | Fair comparison; stable training |
| SAHP | Heads $\in \{2, 4, 8\}$; $H \in \{96, 128, 192\}$; Layers $\in \{1, 2\}$; Dropout $\in \{0, 0.1\}$ | Heads $= 4$, $H = 128$, Layers $= 2$ | Competitive |
| THP | Heads $\in \{2, 4, 8\}$; $H \in \{96, 128, 192\}$; Layers $\in \{1, 2\}$; Dropout $\in \{0, 0.1\}$ | Heads $= 4$, $H = 128$, Layers $= 2$ | As above |
| S4/Mamba | $H \in \{192, 256\}$; $B \in \{50, 100, 200\}$; LR $\in \{1\times10^{-3}, 2\times10^{-3}\}$ | $H = 256$, $B = 100$, LR $1\times10^{-3}$ | Accuracy/cost balance; Table 13 |
| Quadrature $n_q$ | $\{4, 8\}$ | 8 (LOBSTER/MIMIC); 4 (FX) | Shorter horizons in FX |
| Temp. scaling | bins $\in \{10, 20, 50\}$; optimizer=LBFGS (10 steps) | ECE@20 reported; LBFGS | Standard practice for reliability |
| Ensembles | $n \in \{3, 5\}$; seed $\in \{0, \ldots\}$ | $n = 5$ | Improves PIT/KS and coverage |
| CQR (Supp.) | $(\tau_\ell, \tau_u) = (0.05, 0.95)$; calibration quantile | $\alpha = 0.1$; direct CQR | Valid 90% PIs by design |