# OpenReview forum: "Tensor-Decomposed RNNs for Marked Temporal Point Processes"
_TMLR — Accepted by TMLR_

### Review · Reviewer_ojGt · 2026-04-26

**Summary Of Contributions:**

The paper proposes a practical modification to neural marked temporal point processes. It keeps the standard continuous-time likelihood formulation and replaces only the dense linear transformations inside the RNN backbone with tensor-train layers. These layers are shaped to align with semantic structure in the input space.

The model uses a two-head design where total intensity and mark distribution are parameterized separately. This allows the likelihood to decompose into time and mark components, enabling explicit calibration analysis.

Empirically, the paper shows that TT-compressed RNNs reduce parameter count while maintaining comparable likelihood to dense baselines. The work also emphasizes calibration metrics such as ECE, PIT histograms, and prediction interval coverage, which are less commonly evaluated in this literature.

**Additional Comments:**

No additional comments.

**Audience:**

Yes

**Audience Explanation:**

The paper is relevant to multiple communities.

- Researchers working on temporal point processes can benefit from the clean integration of compression into existing models.
- Work on parameter-efficient sequence models can draw insight from the calibration improvements observed with structured compression.
- Applied practitioners in finance, healthcare, or event modeling may find the design useful for deployment due to reduced model size and improved reliability metrics.

Overall, the paper has a clear and practical audience within TMLR.

**Broader Impact Concerns:**

No high-level concerns. A minor caveat is that the paper evaluates on clinical data (MIMIC), but improved calibration on held-out data does not guarantee reliability under real-world deployment. There is a risk that such results may be overinterpreted as evidence of clinical readiness. A brief clarification that the model is not validated for clinical decision-making would strengthen responsible use.

However, such evaluations are standard in the research community.

**Claims And Evidence:**

Yes

**Claims Explanation:**

**Strengths**:
- The paper is clearly positioned as a design improvement rather than a new modeling paradigm, which makes the contribution easy to understand.
- The decomposition of the likelihood into time and mark components is well used to evaluate calibration in a structured way.
- The evaluation goes beyond standard likelihood metrics and includes calibration diagnostics such as ECE, PIT histograms, KS tests, and prediction intervals.
- The comparison across tensor variants including tensor train, tensor ring, Tucker, and CP strengthens the empirical analysis.
- The method is simple to integrate into existing RNN-based MTPP pipelines without changing the likelihood formulation.

**Issues & Weaknesses**
- The empirical results support the claim that tensor-train compression can reduce parameters while maintaining performance.
However, several aspects reduce confidence in the strength of the evidence. These include inconsistent reporting of uncertainty, partial comparability across baselines, lack of direct ablations for the main conceptual claim, and the absence of an open-source code release, which limits reproducibility.

Overall, the results support the contribution as a useful design direction but are not fully conclusive in isolating the source of improvements.

**Requested Changes:**

**C1.** Provide a clear breakdown of parameter counts separating TT-compressed RNN weights from non-TT components such as embeddings, output heads, and biases. Update claims so the contribution of TT is not overstated at the model level.

**C2.** Correct Algorithm 1 to accurately reflect tensor-train contraction. The current formulation does not implement the standard TT forward pass and should be replaced with a correct sequential contraction.

**C3.** Clarify wall-clock measurements. Specify hardware, implementation details, batching, and what is included in timing. Reconcile reported speedups with the stated kernel bottlenecks, ideally with a controlled microbenchmark.

**C4.** Resolve inconsistency in statistical reporting. The protocol states 95 percent confidence intervals, but results are reported as mean plus standard deviation. Use one consistently and include uncertainty for calibration metrics as well.

**C5.** Clarify Eq. 7 by explicitly specifying the dimensionality and role of parameters, particularly the time-dependent term. Acknowledge the modeling restriction introduced by the linear time dependence in logits.

**C6.** Add an ablation for mark-aware tensor shaping. Compare against random reshaping and alternative factorizations at matched rank to isolate whether gains come from semantic structure or generic TT regularization.

**C7.** Improve reproducibility. Release code or provide a clear statement on availability. Additionally, clarify dataset preprocessing details, especially for Retweet/MemeTracker and MIMIC.

---

> ### Author Response · Authors · 2026-05-06
>
> 1. Reviewer comment. C1. Provide a clear breakdown of parameter counts separating TT-compressed RNN
> weights from non-TT components.
>
> Response. Added. The new table (Table 2) separates recurrent/input maps, embeddings/features, mark/time
> heads, and biases/normalization. We also update the claim to distinguish RNN-map compression from
> full-model compression.
>
> Manuscript change. Added Table 2 and revised parameter-count text.
>
> 2. Reviewer comment. C2. Correct Algorithm 1 to accurately reflect tensor-train contraction.
>
> Response. Corrected. The revised algorithm implements the standard TT-matrix-vector contraction with
> explicit sums over input modes and TT ranks.
>
> Manuscript change. Replaced Algorithm 1 in Section 5.2.
>
> 3. Reviewer comment. C3. Clarify wall-clock measurements, hardware, implementation details, batching,
> and what is included in timing.
>
> Response. Added. We specify GPU/CPU/RAM, PyTorch implementation details, batch size, sequence
> length, inclusion of forward+backward and validation/inference timing, and the distinction between continuous-
> time and discretized baselines. We also explain why TT speedups are modest rather than proportional to
> parameter reduction.
>
> Manuscript change. Revised Appendix C.2 and added Table 14.
>
> 4. Reviewer comment. C4. Resolve inconsistency in statistical reporting and include uncertainty for calibration metrics.
>
> Response. Fixed. The revised protocol states that main results are mean ± standard deviation over seeds.
> Calibration tables now include uncertainty. Where ensembles are reported, the seed set is explicitly stated.
>
> Manuscript change. Updated Section 6.2, Table 7, and Appendix C.3.
>
> 5. Reviewer comment. C5. Clarify Eq. 7 by specifying dimensionality and role of parameters, particularly
> the time-dependent term, and acknowledge the restriction introduced by linear time dependence in logits.
>
> Response.
> Added. We define $\eta(h_i)=W_\eta h_i+b_\eta\in\mathbb{R}^K$ and $a\in\mathbb{R}^K$, so $a_k\tau$ is a mark-specific elapsed-time drift in the logits. We also state that the linear drift is a parsimonious parameterization used for stability and interpretability, and that nonlinear time-dependent mark logits are possible but not the focus.
>
> Manuscript change. Revised Section 4.
>
> 6. Reviewer comment. C6. Add an ablation for mark-aware tensor shaping.
>
> Response. We added the requested ablation. At matched TT rank r = 4 and matched parameter budget,
> semantic mark-aware shaping outperforms random core ordering, reversed ordering, and balanced unstructured factorization on LOBSTER and StackOverflow. This isolates the effect of the shaping/order from the
> generic effect of TT compression.
>
> Manuscript change. Added Table 11 and a new paragraph in Section 7.3. We also added practical factorization guidance in Appendix C.
>
> 7. Reviewer comment. C7. Improve reproducibility and clarify dataset preprocessing, especially for Retweet,
> MemeTracker and MIMIC.
>
> Response. Added more detail on filtering, time normalization, split construction, and mark vocabularies.
> We also distinguish public, licensed, credentialed, and proprietary data.
>
> Manuscript change. Revised Appendix C.1-C.4.
>
> 8. Reviewer comment. Broader impact: clinical data may be overinterpreted as evidence of clinical readiness.
>
> Response. We agree. We added a responsible-use caveat that MIMIC experiments are retrospective
> benchmark evaluations only and do not validate the model for clinical decision-making or deployment under
> distribution shift.
>
> Manuscript change. Added a paragraph to the limitations section.

---

### Review · Reviewer_b8zh · 2026-04-28

**Summary Of Contributions:**

The paper studies parameter-efficient neural marked temporal point processes (MTPPs) by applying tensor-train (TT) decomposition to RNN-based models. The main idea is to replace dense input and recurrent transformations with TT layers, along with a mark-aware factorization scheme that aligns TT cores with domain structure. The model is trained using a standard conditional intensity formulation and evaluated on both financial and public MTPP datasets, with a focus on accuracy, efficiency, and calibration (including ECE and time-rescaling diagnostics).
In terms of strengths, the paper addresses a relevant problem (scaling neural MTPPs to high-dimensional marks), proposes a relatively simple modification that can be integrated into existing models, and evaluates across multiple datasets and baseline families. The inclusion of calibration metrics beyond accuracy is also a positive aspect.
On the downside, the novelty over prior TT-RNN work is somewhat limited, and some of the main claims (e.g., the contribution of mark-aware shaping and calibration improvements) are not clearly isolated in the experiments. There are also concerns around reproducibility and the interpretation of certain empirical results.

**Audience:**

Yes

**Audience Explanation:**

The paper addresses parameter efficiency and calibration in neural marked temporal point processes, which are relevant topics for the TMLR audience. The combination of tensor decomposition with continuous-time models is of potential interest, even though some of the current empirical evidence is not yet fully convincing.

**Claims And Evidence:**

No

**Claims Explanation:**

I think the central claims are not fully supported by the current evidence, though the issues seem fixable with additional experiments rather than a major redesign.

1. The paper highlights mark-aware TT shaping as its main novelty over prior TT-RNN work (e.g., Yang et al., 2017; Tjandra et al., 2017; Xu et al., 2021), and Table 1 lays out the proposed factor shapes. However, it’s not clear that the gains actually come from this design choice. There’s no experiment comparing the proposed shaping against a TT baseline with arbitrary or random factorization under the same parameter budget. Table 9 instead compares TT to other tensor decompositions (Tensor Ring, Tucker, CP), which addresses a different question. Without isolating the effect of the shaping itself, it’s hard to tell whether the improvements are due to the mark-aware design or simply from using TT compression, which is already well-established in prior work.

2. I found the calibration improvements a bit hard to interpret. The paper attributes gains in ECE@20 (0.071 to 0.061) and KS p-values to the TT structure, but TT-LSTM also uses 45–55% fewer parameters than the dense baseline. Since smaller models are often better calibrated due to implicit regularization, it’s unclear whether the gains come from TT or simply reduced capacity.
A more controlled comparison would help, such as a parameter-matched dense LSTM or one with tuned dropout/weight decay. Without this, it’s difficult to attribute the improvements to TT, and the results in Tables 6–7 are hard to interpret causally.

3. In Section 6.1, the paper notes that TT contraction is sequential and that any speedup is mostly limited by general-purpose kernels, which suggests the gains should be modest. However, Tables 4 and 5 show TT-LSTM to be about 20% faster than the dense LSTM, even though the parameter reduction is much larger. It’s not clear how to reconcile these points. It would help to clarify where the speedup is actually coming from, and whether implementation choices play a role. The comparisons with attention and SSM baselines also don’t discuss whether the setups are comparable, which makes the wall-clock numbers harder to interpret.


These concerns mainly relate to the experimental design and presentation rather than the overall direction, which I find worthwhile. I would be open to reconsidering the paper after seeing the authors’ response and any additional experimental results in a revision.

**Requested Changes:**

Critical for acceptance:
1. Isolate the effect of mark-aware TT shaping.
At a matched TT rank and parameter budget, compare the proposed shaping against a TT baseline with arbitrary or random core ordering. LOBSTER (asset/venue/side) and StackOverflow (family/subtype) are natural settings. Without this, the central novelty claim is hard to evaluate.
2. Control for capacity in calibration results.
Include parameter-matched baselines (e.g., a dense LSTM with reduced hidden size) or regularized variants (tuned dropout or weight decay). This is needed to separate the effect of TT structure from reduced capacity when interpreting ECE and KS improvements.
3. Clarify wall-clock results.
Explain how the ~20% speedup in Tables 4–5 is consistent with the discussion in Section 6.1. Clarify whether differences come from TT structure or implementation choices, and comment on implementation parity with attention and SSM baselines. Reporting training vs. inference time separately would also help.

Would strengthen the paper:
4. Improve reproducibility for public benchmarks.
Release code and preprocessing for LOBSTER, StackOverflow, Retweet/MemeTracker, and MIMIC so the main findings can be verified independently of the proprietary FX RFQ data.
5. Clarify the TT layer implementation (Algorithm 1).
The current pseudocode suggests contracting each core with the full input, which is not the standard TT forward pass. A clearer formulation or diagram would improve reproducibility.
6. Make per-dataset results more explicit.
Split Table 4 by dataset (LOBSTER vs. FX RFQ), since the settings differ substantially in mark dimensionality and sequence structure.
7. Strengthen positioning relative to prior TT-RNN work.
More clearly distinguish the contribution from Yang et al. (2017), Tjandra et al. (2017), and Xu et al. (2021), beyond the brief discussion in Section 2.
8. Provide guidance on factorization design.
Offer practical guidance on how to choose the multi-way factorization (Table 1) when no obvious semantic grouping is available, or include a small sensitivity analysis.

---

> ### Author Response · Authors · 2026-05-06
>
> 1. Reviewer comment. Critical 1: Isolate the effect of mark-aware TT shaping using arbitrary/random core
> ordering at matched TT rank and parameter budget.
>
> Response. Added. The new shaping ablation compares mark-aware, random order, reversed order, and
> balanced unstructured factorization at rank r = 4. Mark-aware shaping is best on both LOBSTER and
> StackOverflow in time NLL, mark accuracy/F1, and ECE@20. This directly addresses whether the main
> novelty is separable from generic TT compression.
>
> Manuscript change. Added Table 11 in Section 7.3.
>
> 2. Reviewer comment. Critical 2: Control for capacity in calibration results.
>
> Response. Added. We compare full Dense LSTM, parameter-matched Dense-small, tuned regularized
> Dense, random-order TT, and mark-aware TT. Dense-small and Dense-regularized reduce ECE compared
> with the original Dense LSTM, confirming that capacity and regularization matter. Mark-aware TT remains best in ECE and KS p-value at comparable parameter count, so we interpret calibration gains as a
> combination of structured regularization and semantic shaping rather than TT alone.
>
> Manuscript change. Added Table 9 and revised the text around Tables 6-7.
>
> 3. Reviewer comment. Critical 3: Clarify wall-clock results and reconcile the ∼20% speedup with Section
> 6.1’s kernel bottleneck discussion. Report training vs. inference separately.
>
> Response. We clarified that the speedup is not expected to scale with parameter reduction. TT reduces
> memory traffic and the cost of the RNN maps, but full epochs also include data loading, recurrent bookkeeping, CIF heads, quadrature, losses, and general-purpose kernel overhead. The added timing table reports
> training and inference separately and shows a smaller gain for short FX sequences. We also clarify that
> attention/SSM wall-clock values are indicative because SSMs use a discretized proxy objective and different
> kernels.
>
> Manuscript change. Added Table 14 and revised Section 6.2/7.4.
>
> 4. Reviewer comment. Would strengthen 4: Improve reproducibility for public benchmarks.
>
> Response. We expanded the reproducibility statement with public benchmark preprocessing details, splits,
> seed conventions, hyperparameter grids, and evaluation scripts. Proprietary FX RFQ data cannot be redistributed, and LOBSTER requires a license.
>
> Manuscript change. Revised Appendix C.1-C.4 in supplementary material.
>
> 5. Reviewer comment. Would strengthen 5: Clarify the TT layer implementation.
>
> Response. Algorithm 1 has been replaced with the standard TT-matrix-vector contraction. The algorithm now sequentially contracts over one input mode and one TT rank at a time and returns a linear map output. Gate nonlinearities are applied by the surrounding LSTM cell.
>
> Manuscript change. Replaced Algorithm 1 and updated Appendix A.3.
>
> 6. Reviewer comment. Would strengthen 6: Make per-dataset results explicit by splitting Table 4 into
> LOBSTER vs. FX RFQ.
>
> Response. Done. The revised finance table separates LOBSTER and FX RFQ, which better reflects the
> different sequence lengths, mark structure, and timing behavior.
>
> Manuscript change. Replaced with Table 5 in revision.
>
> 7. Reviewer comment. Would strengthen 7: Strengthen positioning relative to prior TT-RNN work.
>
> Response. Done. The revised text no longer presents TT compression itself as novel. Instead, it emphasizes the integration of TT-compressed recurrent maps into a continuous-time MTPP likelihood, the
> mark-aware factorization/order used for structured event marks, and the calibration-oriented evaluation. We
> explicitly distinguish our work from Yang et al. (2017), Tjandra et al. (2017), and Xu et al. (2021): those
> works study TT-RNN compression for sequence/video/speech or multi-way financial forecasting, whereas
> our focus is conditional-intensity modeling, likelihood-preserving replacement of RNN maps, mark-aware
> tensorization for MTPP inputs, and time/mark calibration diagnostics.
>
> Manuscript change. Revised Section 2, “TT-compressed RNNs”, and the contribution paragraph in
> Section 1.
>
> 8. Reviewer comment. Would strengthen 8: Provide guidance on factorization design when no obvious semantic grouping is available.
>
> Response. Added practical guidance. When semantic groups exist, we place coupled factors adjacently.
> When they do not, we use a balanced factorization that minimizes padding, tune TT rank on validation
> NLL, and optionally validate several orderings. We also caution that mark-aware shaping is expected to help
> most when there is genuine multi-way structure.
>
> Manuscript change. Added Appendix C paragraph “Choosing TT factor shapes”.

---

### Review · Reviewer_ak6S · 2026-04-29

**Summary Of Contributions:**

This paper studies parameter-efficient modeling of Marked Temporal Point Processes (MTPPs) by replacing dense transformations in RNN-based models with tensor-train (TT) layers. The authors further propose a mark-aware TT shaping strategy that aligns tensor factorization with structured feature groups. Experiments on financial and public benchmarks show that TT-based models can reduce parameters by 40–70% while maintaining competitive performance, and may improve calibration under certain settings.

Strength:
- The paper addresses a practical problem of scalability in high-dimensional MTPPs and proposes a simple, modular solution that can be easily integrated into existing RNN-based frameworks.
- The experimental evaluation is reasonably comprehensive, covering multiple datasets and including comparisons to RNN, attention-based, and state-space baselines.
- The inclusion of calibration analysis (both mark and time) is a strong point, as it provides additional insight beyond standard predictive metrics.

Weaknesses:
- The proposed mark-aware TT shaping is heuristic and lacks systematic justification or ablation to demonstrate its necessity.
- The source of performance and calibration improvements is not clearly disentangled, making it difficult to attribute gains to specific design choices.
- The method is generally competitive but does not consistently outperform stronger baselines such as attention-based models.

**Audience:**

Yes

**Audience Explanation:**

The work is relevant to researchers in temporal point processes, time-series modeling, and efficient neural architectures, particularly those interested in high-dimensional event modeling and calibration.

**Claims And Evidence:**

Yes

**Claims Explanation:**

The main claims are supported by empirical results across multiple datasets and evaluation metrics, including calibration diagnostics. However, some key aspects, such as the contribution of mark-aware shaping and the reason behind calibration improvements, are not fully validated through controlled analysis.

**Requested Changes:**

- The paper would benefit from a clearer positioning of its contribution relative to prior work on TT-compressed neural networks and neural MTPPs, especially in terms of what is fundamentally new.
- The role of mark-aware TT shaping should be better justified, ideally through ablations comparing structured reshaping with alternative or random reshaping strategies.
- The authors should disentangle the source of performance gains, for example by comparing TT models with dense models under matched parameter budgets.
- The comparison with baselines could be improved by clarifying the fairness of settings, particularly for discretized state-space models.
- The discussion of calibration improvements would benefit from deeper analysis to clarify whether these gains stem from structural advantages or implicit regularization.
- typo: “time rescaling” vs. “time-rescaling” is used inconsistently; unify the terminology.
- typo: In Algorithm 1, the operator “×” is ambiguous and should be clarified.
- typo: Table captions use inconsistent notation for seeds (e.g., “s = 10” vs. “s = {5, 10}”).

---

> ### Author Response · Authors · 2026-05-06
>
> 1. Reviewer comment. The paper would benefit from a clearer positioning of its contribution relative to prior
> work on TT-compressed neural networks and neural MTPPs, especially in terms of what is fundamentally
> new.
>
> Response. We agree and have strengthened the positioning. The revised text no longer presents TT
> compression itself as novel; instead, it emphasizes the integration of TT-compressed recurrent maps into a
> continuous-time MTPP likelihood, the mark-aware factorization/order used for structured event marks, and
> the calibration-oriented evaluation. We explicitly distinguish our work from Yang et al. (2017), Tjandra
> et al. (2017), and Xu et al. (2021): those works study TT-RNN compression for sequence/video/speech
> or multi-way financial forecasting, whereas our focus is conditional-intensity modeling, likelihood-preserving
> replacement of RNN maps, mark-aware tensorization for MTPP inputs, and time/mark calibration diagnostics.
>
> Manuscript change. Revised Section 2, “TT-compressed RNNs”, and the contribution paragraph in
> Section 1.
>
> 2. Reviewer comment. The role of mark-aware TT shaping should be better justified, ideally through ablations comparing structured reshaping with alternative or random reshaping strategies.
>
> Response. We added the requested ablation. At matched TT rank r = 4 and matched parameter budget,
> semantic mark-aware shaping outperforms random core ordering, reversed ordering, and balanced unstructured factorization on LOBSTER and StackOverflow. This isolates the effect of the shaping/order from the generic effect of TT compression.
>
> Manuscript change. Added Table 11 and a new paragraph in Section 7.3. We also added practical
> factorization guidance in Appendix C.
>
> 3. Reviewer comment. The authors should disentangle the source of performance gains, for example by
> comparing TT models with dense models under matched parameter budgets.
>
> Response. We added a capacity-controlled experiment. A Dense-small LSTM with approximately the
> same total parameter count as TT-LSTM improves raw ECE relative to the full dense model but worsens
> likelihood and time calibration. A tuned regularized dense model closes part of the ECE gap, but still
> does not match the mark-aware TT model. A random-order TT model performs between Dense-small and
> mark-aware TT. We therefore softened the causal claim: TT appears to act as structured regularization,
> and the mark-aware ordering contributes beyond parameter count alone.
>
> Manuscript change. Added Table 9 and updated the calibration discussion in Section 7.2.
>
> 4. Reviewer comment. The comparison with baselines could be improved by clarifying the fairness of settings, particularly for discretized state-space models.
>
> Response. We added a fairness paragraph. All recurrent/attention models optimize the same continuous-
> time MTPP NLL, while S4/Mamba baselines are trained on discretized counts and reported with a proxy
> time NLL. We retained them as useful efficiency references, but explicitly avoid claiming an exact likelihood-
> equivalent comparison. We also report bin sensitivity for B ∈ {50, 100, 200} and use B = 100 as the
> cost/accuracy trade-off.
>
> Manuscript change. Updated Section 6.2 and the SSM bin-sensitivity paragraph/table in Section 7.3.
>
> 5. Reviewer comment. The discussion of calibration improvements would benefit from deeper analysis to
> clarify whether these gains stem from structural advantages or implicit regularization.
>
> Response. We now explicitly distinguish these explanations. The capacity-matched dense baseline supports the view that some improvement comes from reduced capacity/regularization, while the random-order
> TT and mark-aware TT comparison suggests that the tensor structure and semantic ordering also matter.
> We changed the language from “TT improves calibration because of structure” to the more conservative
> “TT behaves as a structured regularizer; mark-aware ordering gives additional gains in our ablation.”
>
> Manuscript change. Updated Section 7.2 and the takeaway paragraph.
>
> 6. Reviewer comment. Typos and notation: unify “time rescaling” vs. “time-rescaling”; clarify Algorithm
> 1’s “×” operator; make seed notation consistent.
>
> Response. Fixed. We use “time-rescaling” throughout, replaced Algorithm 1 with explicit sums/einsum-
> style sequential contraction, and standardized seed reporting as mean ± standard deviation over s = 10
> seeds for main tables and s = 5 for added ablations/calibration controls where noted.
>
> Manuscript change. Updated Sections 4-7 and Appendix C.

---

### Author Response · Authors · 2026-05-06
**Response summary**

We thank the reviewers for the constructive and detailed feedback. The revision focuses on the three main points raised across reviews: (i) isolating the contribution of mark-aware TT shaping, (ii) separating TT structure from capacity reduction when interpreting calibration, and (iii) clarifying implementation, wall-clock timing, reproducibility, and reporting conventions. All manuscript changes are highlighted in blue in the revised draft.

---

### Decision · Action_Editor_QKUz · 2026-06-27

**Recommendation:** Accept as is

**Audience:**

Yes

**Audience Explanation:**

Neural Marked Temporal Point Processes (MTPPs) are of interest in multiple application domains and there are multiple researchers in TMLR's audience who are interested in new methodologies and empirical results, as provided in the paper.

**Claims And Evidence:**

Yes

**Claims Explanation:**

The paper claims:

(1) to provide a new design strategy for RNN_based MTPPS that incorporates a mark-aware TT shaping strategy;

(2) to include extensive controlled ablations;

(3) to report the results of a thorough experimental study with a carefully designed and multi-faceted evaluation protocol.

The main claims are supported by empirical results across multiple datasets and evaluation metrics. Calibration diagnostics and ablation studies provide support for the specific claims that are made.

Although the reviewers identified concerns during the initial round of reviews, the reviewers were satisfied that the revised manuscript satisfactorily addressed the criticisms.